# Management and 1-Year Outcome in Elderly Patients with Hip Fracture Surgery Receiving Anticoagulation (Warfarin or DOAc) or P2Y12 Antiplatelet Agents

**DOI:** 10.3390/jcm12196178

**Published:** 2023-09-25

**Authors:** Carlo Rostagno, Gaia Rubbieri, Mattia Zeppa, Alessandro Cartei, Alice Ceccofiglio, Giulio Maria Mannarino, Gualtiero Palareti, Elvira Grandone

**Affiliations:** 1Dipartimento Medicina Sperimentale e Clinica, Università di Firenze, 50134 Firenze, Italy; mattia.zeppa@stud.unifi.it; 2Medicina Interna e Post-Chirurgica, AOU Careggi, 50134 Firenze, Italy; rubbierig@aou-careggi.toscana.it (G.R.); carteia@aou-careggi.toscana.it (A.C.); ceccofiglioa@aou-careggi.toscana.it (A.C.); mannarinog@aou-careggi.toscana.it (G.M.M.); 3Fondazione Arianna Anticoagulazione, 40138 Bologna, Italy; gualtiero.palareti@unibo.it; 4Thrombosis and Haemostasis Unit, Fondazione I.R.C.C.S. “Casa Sollievo della Sofferenza”, 71013 San Giovanni Rotondo, Italy; elvira.grandone@unifg.it

**Keywords:** DOAc, hip fracture, P2Y12 inhibitors, warfarin

## Abstract

(1) Background: Little prospective data exist regarding the perioperative management and long-term prognosis of elderly patients receiving treatment with antithrombotic drugs and undergoing urgent surgery for a hip fracture. (2) Methods: The study included patients who required hip surgery and were receiving warfarin, DOAc or P2Y12 antiplatelet agents at the moment of trauma. Ongoing antithrombotic treatment was managed according to existing recommendations. The endpoints of the study were the time to surgery, perioperative bleeding, the need for transfusion and, finally, mortality, major cardiovascular events and re-hospitalization at 6 and 12 months. (3) Results: The study included a total of 138 patients. The mean age was 86 years; 75.4% were female. Eighty-two received DOAc, thirty-six received warfarin and twenty received P2Y12 inhibitors. The controls were 283 age- and sex-matched patients who did not receive antithrombotic treatment. A total of 38% of patients receiving warfarin underwent surgery <48 h, 52% receiving DOAc, 55% receiving P2Y12 inhibitors and, finally, 82% in the control group. Perioperative bleeding and the need for transfusion were not different between the four groups. Mortality at 6 months was higher in patients receiving warfarin and P2Y12 inhibitors (30% and 25%) in comparison to DOAc and the control group (11.6% and 10% *p* < 0.0001). Similarly, the other endpoints were more frequent in patients receiving warfarin and P2Y12 inhibitors. The trend was maintained for 12 months. No significant differences in mortality were found between early (<48 h) and late (>48 h) surgery independent of the type of treatment. (4) Conclusions: Our study confirmed that anticoagulants delay surgery in patients with hip fractures; however, intervention > 48 h is not associated with a poorer prognosis. This finding is relevant as it underlines that, in patients at high risk of postoperative cardiovascular complications, the careful management of anticoagulation before surgery may compensate for the delay of surgery with a very low in-hospital mortality rate (<1%). One-year survival was significantly lower in patients receiving warfarin, probably related to their worse risk profile at the moment of trauma survival.

## 1. Introduction

Fragility fractures are a major healthcare problem worldwide. Due to increasing life expectancy and associated demographic changes, the incidence of fractures and post-fracture disability is constantly increasing [1]. With hip fractures, early surgery within 24–48 h of trauma is associated with better outcomes both in terms of the recovery of functional capacity and overall early and 1-year survival [2,3]. In the last decade, moreover, several studies have demonstrated that a multidisciplinary approach in hip fracture patients leads to a decrease in hospital complications and a decrease in the postoperative rate of dependence, as well as a functional decline, other than a reduction in early and late mortality [4,5]. It is still debated whether general or neuraxial anesthesia should be preferred in elderly frail patients. Neuman et al. [6] did not determine any advantages of neuraxial in comparison to general anesthesia in older adults concerning survival of and recovery from ambulation after 60 days. The incidence of postoperative delirium was similar to the two types of anesthesia. Neuraxial anesthesia is related to a limited higher risk of hypotension that may be controlled by norepinephrine administration [7].

Aging is strongly associated with multimorbidity and polypharmacy [8]. No less than 30% of patients receive antithrombotic therapy (DOAc, warfarin or P2Y12 antiplatelet inhibitors) at the moment of trauma [5,6]. The perioperative management of oral anticoagulant therapy in elderly patients who require urgent orthopedic surgery for hip fractures is challenging. In the last decade, DOAc has been extensively used in the replacement of warfarin in patients with non-valvular atrial fibrillation; however, their handling in the perioperative period and long-term effects in patients who need non-cardiac surgery require further studies. Furthermore, the indications of thrombo-prophylaxis in patients with mechanical heart valves are far from being established [9,10]. Several papers have reported a significant delay in the time to surgery in patients with hip fractures treated with warfarin or DOAc, associated with an increase in the length of hospital stay, the risk of postoperative complications and decreased survival [11,12,13,14,15]. To date, there has been no consensus on the timing of surgery after the discontinuation of anticoagulant treatment nor whether P2Y12 antiplatelets should be withheld before surgery [16,17]. The time of resumption of therapy as well as “bridge” treatment in patients with warfarin remains a matter for debate. Moreover, many centers use general anesthesia, while in our center, as well as in several centers in northern Europe, neuraxial anesthesia is the preferred technique. Both anticoagulants and P2Y12 inhibitors significantly increase the risk of hemorrhagic complications of neuraxial anesthesia, and surgery should be deferred until the recovery of hemostatic competence. The choice of general anesthesia may allow for surgery without the withdrawal of P2Y12 inhibitors and decrease the delay to surgery by at least 24 h in patients receiving DOAc. Finally, no data exist regarding the long-term outcomes of these patients after surgery. We report on the results from a prospective, observational study investigating the management of patients receiving DOAc, warfarin, or P2Y12 antiplatelet inhibitors that need urgent intervention for hip fractures. We previously demonstrated that the stabilization of concomitant comorbidities and integrated perioperative clinical management may safely decrease the time to surgery and shorten hospital stays in frail patients with hip fractures [18]. In the present study, we prospectively evaluate data on the peri-operative management of antithrombotic therapies in elderly patients who need urgent orthopedic surgery. The endpoints of the study are the time to surgery, perioperative bleeding, the need for transfusion and, finally, mortality, major cardiovascular events and re-hospitalization at 6 and 12 months.

## 2. Materials and Methods

### 2.1. Study Design

This study was an observational, prospective cohort study designed to investigate perioperative management of patients treated with DOAc, warfarin or P2Y12 antiplatelet inhibitors undergoing urgent orthopedic surgery for fragility hip fractures. The study aimed to evaluate the time to surgery, bleeding and the transfusion rate in hospital and the 12-month mortality rate according to the antithrombotic treatment at the moment of trauma (warfarin, DOAc, P2Y12 inhibitors). The study was part of a project of the Italian Health Ministry and Regione Toscana–RF-2010-2316600–and was approved by the Ethical Committee Area Vasta Toscana Centro on 19 February 2019 (n°6590/2109). All patients gave written informed consent to treatment and clinical data for research purposes were collected during admission. The study was conducted according to the Declaration of Helsinki.

### 2.2. Study Population

The study was conducted at a third-level teaching hospital. The inclusion criteria were patients aged >70 years, able to give informed consent, and undergoing anticoagulant-antiplatelet treatment at the time of trauma. The exclusion criteria were age <70 years, refusal of follow-up, patients with active cancer and patients in double antiplatelet therapy for recent (<6 months) coronary revascularization. Data were collected between April 2019 and July 2021. All patients underwent multidisciplinary evaluation according to previously reported protocol [12]. Anticoagulants were withdrawn during hospital admission. In patients treated with DOAc, surgery was scheduled according to EHRA guidelines [19]. As a control group, we considered hip fracture patients referred in the same period who were not in antithrombotic therapy, including those receiving low-dose aspirin, since it did not affect the surgical schedule.

Prophylactic LMWH was started within 12 h after surgery and anticoagulation was post-operatively resumed on days 3–4 according to local bleeding. In patients receiving warfarin, vitamin K was administered at admission in low-risk patients. Prophylactic LMWH was started when INR was <2. Bridge treatment was limited to high-risk patients (mechanical heart valves, recent systemic or pulmonary thromboembolism). Warfarin was usually resumed on postoperative day 2 and LMWH was withdrawn when INR reached therapeutic range. Finally, treatment with P2Y12 inhibitors was not stopped before surgery and the drugs were only not administered on the day of surgery.

The demographic data were collected in an electronic database, along with the indication of antithrombotic treatment, comorbidities, the time to surgery, the need for and the number of RBC transfusions, pre- and postoperative hemoglobin (Hb), platelets, creatinine, INR and aPTT. Transfusion was indicated for Hb levels <8 mg/dL. Above these values, the decision was left to each physician according to the clinical conditions of the patient.

A telephone follow-up was scheduled to investigate mortality, major cardiovascular events (heart failure, acute myocardial infarction, stroke) and re-hospitalization at 6 and 12 months after surgery.

### 2.3. Statistical Analysis

The parameters considered were reported as mean values and standard deviations. Categorical variables were expressed as distribution frequencies. In the comparison between groups, the Student’s test for unpaired data and continuous parameters was used, and the χ^2^, the Fisher’s exact test, and ANOVA were used for categorical variables. Survival analysis was performed using two-tailed Kaplan-Meier curves. The differences between groups were compared using the Log-Rank test. A probability value of 0.05 was considered statistically significant. The statistical analysis was performed with Stata 12.0 software (StataCorp. 2019. Stata Statistical Software: Release 16. College Station, TX, USA: StataCorp LLC.).

## 3. Results

In the study, 138 patients were referred for hip fractures (50 for the neck of the femur, 60 per trochanteric and 12 for subtrochanteric). The mean age was 86 years; 104 (75,4%) were female. In 29%, at least four activities during the basic activity of daily living (BADL) were lost before hospitalization. In addition, 283 age- and sex-matched patients who were not receiving antithrombotic treatment were considered as a control group. The 90 patients with aspirin were included in the control group. A separate sub-analysis did not show clinical differences nor different outcomes between patients with no antithrombotic treatment and those receiving low-dose aspirin. A comparison of the characteristics of patients and the control group is reported in Table 1.

Overall, we did not find significant differences regarding age, gender and functional capacity between patients receiving antithrombotic treatment (warfarin, DOAc or P2Y12 inhibitors) and the control group. In patients receiving antithrombotic treatment, the time to surgery was significantly longer (*p* < 0.0001), with only half treated within 48 h from the moment of trauma. Finally, LOS was significantly longer in comparison to the control group.

Among the 138 patients included, 82 (59.4%) were receiving direct oral anticoagulants (22 Dabigatran, 24 Apixaban, 19 Rivaroxaban and 17 Edoxaban), 37 (26.1%) were receiving warfarin and 20 (14.5%) were receiving P2Y12 inhibitors (Figure 1).

Table 2 shows the indications for treatment. Atrial fibrillation was the main indication for therapy in patients receiving DOAc and warfarin. In 75%, P2Y12 inhibitors were used in the primary or secondary prevention of ischemic heart disease or cerebrovascular disease. Among patients on VKA therapy, only 66.4% (20/36 patients) were in the therapeutic range at hospital admission.

The mean time to surgery was 2.95 days: 50% (69 patients) underwent surgery within 48 h of trauma. The overall length of stay (LOS) was 12.6 days. The most frequent complication was delirium, which occurred in 15.9% (22 patients), followed by heart failure and respiratory failure, which occurred in 10% and 5%, respectively. Mortality in patients receiving antithrombotic was 19% (24/138) at 6 months and 25.9% (30/138) at 12 months in comparison to 10% (28/283) and 22% (63/283), respectively, in the control group (*p* < 0.0001). Major cardiovascular events occurred in 20.5% of patients at 6 months (25/138) and in 27% at 12 months (28/138). Finally, the rehospitalization rate was 31.1% (42/138) at 6 months and 40.7% (44/138) at 12 months. Both MACE and rehospitalization were less frequent in the control group.

### 3.1. Analysis of Subgroups of Antithrombotic Treatment

Table 3 reports the clinical characteristics of patients according to treatment at the moment of trauma. There were no significant differences among groups regarding age, gender or the type of fracture.

Major disability (preserved BADL < 4) was more frequent in the DOAc group (39%) than in the warfarin and antiplatelet therapy groups (13% and 15%, respectively, *p* = 0.008).

Among the comorbidities, ischemic heart disease was more frequent in patients on antiplatelet therapy than in patients on DOAc and warfarin therapy.

Warfarin therapy was associated with a significant delay in the time to surgery. Only 38% were treated within 48 h in comparison to 52% receiving DOAc therapy and 55% receiving antiplatelet therapy. In the control group, 82% underwent surgery within 48 h.

In the postoperative period, the decrease in hemoglobin concentration was similar in the four groups, with an average decrease in hemoglobin concentrations of 2 g/dL. Similarly, we did not find any significant difference among the four groups regarding the RBC transfusion requirement. Furthermore, 40% (55/138) of patients in anticoagulation/P2Y12 inhibitors underwent transfusion with at least one unit of concentrated red blood cells vs. 44% (123/283) of the control group.

The rate of postoperative complications was not different across the three groups, with the only exception being the trend of a higher delirium prevalence in patients receiving DOAc.

Mortality at 6 months was higher in patients treated with warfarin and antiplatelet drugs compared to that of patients treated with DOAc (30% and 27% vs. 11.6%, log-rank test *p* < 0.0001) and the controls (Figure 2). Similarly, major cardiovascular events and the need for rehospitalization at 6 months were more frequent in patients receiving warfarin and antiplatelet therapy than in patients receiving DOAc. The trend was also maintained 12 months after surgery (Figure 2). The control group had a significantly lower number of MACE and rehospitalization; however, mortality at 12 months was not significantly different from patients receiving DOAc. There was no difference between patients without any antithrombotic agents and patients treated with low-dose prophylactic aspirin (Figure 3).

### 3.2. Early versus Late Surgery

We evaluated whether a delay in surgery related to antithrombotic surgery influenced early and late mortality. In-hospital mortality was less than 1% (only one patient in DOAC died before discharge). A small but not significant trend toward a decrease in 6- and 12-month mortality was found between patients in the DOAC- and warfarin-treated groups within or after 48 h, while for patients receiving P2Y12 inhibitors, delayed surgery was associated with lower mortality at 6 months (12.5 vs. 36%) (Table 4). No further changes occurred later.

## 4. Discussion

The aim of the study was to evaluate the perioperative management of antithrombotic drugs in patients with hip fractures receiving anticoagulation, with a bleeding rate and the need for blood transfusion. Moreover, we evaluated mortality, cardiovascular complications and re-hospitalization rates at 6- and 12-month follow-ups. Early surgery, within 48 h of the initial trauma, has been reported to improve functional recovery, decrease hospital complications and reduce both early and 12-month mortality [20,21,22,23].

The need for early surgery in patients receiving anticoagulants, however, must be understood as having a high bleeding risk related to “iatrogenic “coagulopathy. In non-anticoagulated subjects, peri-operative blood loss has been calculated to be, on average, 1.86 L in patients who had undergone intramedullary nailing versus 1.30 L treated with arthroplasty [24]. The need for blood transfusion has been reported to be between 40 and 72%, being higher in patients needing long nails. Due to this high bleeding risk, the management of hip fractures should require the restoration of pharmacologically induced clotting abnormalities. Spinal anesthesia is the preferred technique in our hospital and most centers in Europe. It must be emphasized that anticoagulants and P2Y12 inhibitors increase the risk of a spinal or epidural hematoma due to neuraxial anesthesia, and this complication may be catastrophic [25,26]; therefore, surgery should be delayed until the restoration of hemostatic competence or patients are shifted to general anesthesia [27].

With the simple withdrawal of warfarin, when INR is within the therapeutic range, 2.7 to 4.7 days are needed to reach an INR range of 1.2 to 1.6 suitable for surgery, and more for values below 1.3 that allow for safe neuraxial anesthesia. Several schemes of reversal with the administration of vitamin K have been proposed to decrease the delay of surgery. For DOAc-only patients, treatment with dabigatran may cause a prompt reversal of anticoagulation after the administration of idarucizumab; however, in patients with hip fractures, the cost-effectiveness of treatment has never been demonstrated. The antagonist of Xa-inhibitors, andexanet, has still not been indicated for reversal before surgery.

Tran et al. [15], in patients with hip fractures, compared the time to surgery between a group receiving either DOAc or warfarin and an age- and gender-matched control group. This case-control study included 2258 patients. Among the anticoagulated patients, 233 were receiving warfarin and the other 27 were treated with DOAc. Overall, the average time to surgery (TTS) was longer in anticoagulated patients in comparison to the controls (40 h vs. 26.2 h). A longer median TTS was found in the DOAc group compared to the VKA group (66.9 h vs. 39.4 h). There was no difference between the anticoagulated patients and the controls in secondary endpoints, including the rate of stroke, death, bleeding and VTE during admission. Similar results have been reported by other authors [14].

More recently, a trend toward a shorter time to surgery has been reported in patients receiving DOAC treatment; however, this was associated with a higher bleeding rate and the need for transfusion, but not with a different mortality [28,29].

In the present investigation, warfarin was associated with a significant delay in surgery in comparison to DOAc or P2Y12 inhibitor treatment. In the last group, according to present clinical practice, P2Y12 inhibitors were not withdrawn before surgery in 85% of patients. All patients receiving warfarin who had surgery within 48 h had been treated with vitamin K at hospital admission.

Blood losses and the number of blood transfusions were not significantly different in the three groups of patients (warfarin, DOAc and P2Y12 inhibitors). Similarly, the postoperative complications did not significantly differ among groups.

Data regarding long-term survival in patients in antithrombotic treatment after hip surgery are limited. Mortality at 6 and 12 months was higher in patients receiving warfarin. At 6 months in patients receiving P2Y12 inhibitors, mortality was similar to that observed in patients receiving warfarin but, different from warfarin, it did not significantly increase in the following 6 months. In patients receiving DOAc, mortality at 6 and 12 months was significantly lower than in the other two groups and was similar to the general population. Furthermore, 12-month survival was 79%, similar to the controls and analogous to other cohort studies, such as that of a Norwegian study [28]. The subgroup under treatment with DOAc may represent a population at a lower long-term risk of death, among patients in antithrombotic treatment. These results were in line with other studies. Matheron et al. [9] retrospectively analyzed 47 patients with hip fractures receiving DOAc and a control-matched group. Patients receiving DOAc had more comorbidities, delays in surgery and more wound complications in comparison to the controls; however, there were no differences in hemoglobin drop, the need for blood transfusion and mortality. In the study by Lawrence et al. [14], survival analysis showed a higher mortality rate for patients taking warfarin (12-month survival, 66% vs. 76%; hazard ratio, 1.57; 95% CI, 1.21–2.04; *p* < 0.001). In a recent study [29], 7.7% of 535 patients with hip fractures were receiving DOAC at the moment of trauma. The surgical delay was only 3 h longer than for non-anticoagulated patients. The 30-day mortality was 4.9% in the DOAC group in comparison to 5.1% in the controls. The difference may be related to the fact that DOAc treatment is usually administered to a lower-risk population in comparison to warfarin; most patients have non-valvular atrial fibrillation, none have mechanical valve prosthesis, renal function is generally relatively preserved and severe atherosclerotic disease is probably less frequent.

Whether a delay in surgery due to anticoagulation is directly related to increased mortality is still poorly understood. In a recent retrospective study by Noll et al. [30] among 217 patients receiving DOAc, 82 and 135 underwent surgery before and after 48 h, respectively. Mortality at 30 days was 1.2% for early surgery in comparison to 5.9% for delayed surgery (*p* = 0.02), while at 90 days the difference in mortality was not significant (28% and 37.8%, respectively, *p =* 0.07). Mortality in the control group was 4% at 30 days and 35.3% at 90 days. A small but not significant difference was found in our study at 6 and 12 months both for DOAC and warfarin; however, overall mortality was significantly lower than in the paper by Noll (for DOAc, mortality at 6 months was 9.5% and 14% for early and late surgery, respectively).

### Study Limitations

The main limits of the study are the relatively small number of patients included, which may affect the power of statistical analysis, the single-center design of the study and its observational design. We did not find outcome differences related to different surgical treatments (nailing vs. hemi or complete arthroplasty); however, the number of patients included in each group did not allow for a formal statistical analysis. Nevertheless, at variance with most previous investigations, this is a prospective study with well-defined endpoints that allowed some answers to a still controversial topic.

## 5. Conclusions

The lack of consensus and established guidelines on the management of patients with hip fractures receiving anticoagulation, in particular DOAc, highlights the need for randomized controlled trials to establish best practice pathways for the fragile patient population. Our study confirms that anticoagulants delay surgery in patients with hip fractures; however, intervention within >48 h is not associated with a poorer prognosis. This finding is relevant since it underlines that, in patients at high risk of postoperative cardiovascular complications, the careful management of anticoagulation before surgery may compensate for the delay of surgery with a very low in-hospital mortality rate (<1%). The absence of differences in bleeding and transfusion rates among different drugs and controls is consistent with the hypothesis that the careful management of these patients may significantly improve predefined outcomes. Regarding long-term prognosis, we confirmed that patients receiving warfarin have a significantly higher mortality, probably related to a worse risk profile at the moment of trauma.

## Figures and Tables

**Figure 1 jcm-12-06178-f001:**
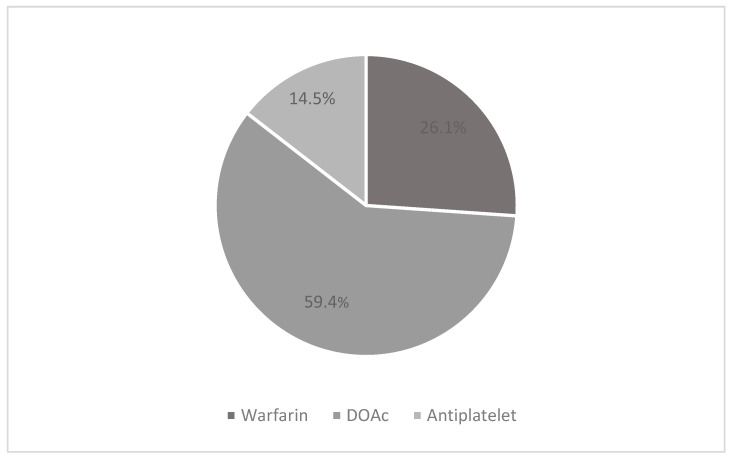
Pharmacological therapy.

**Figure 2 jcm-12-06178-f002:**
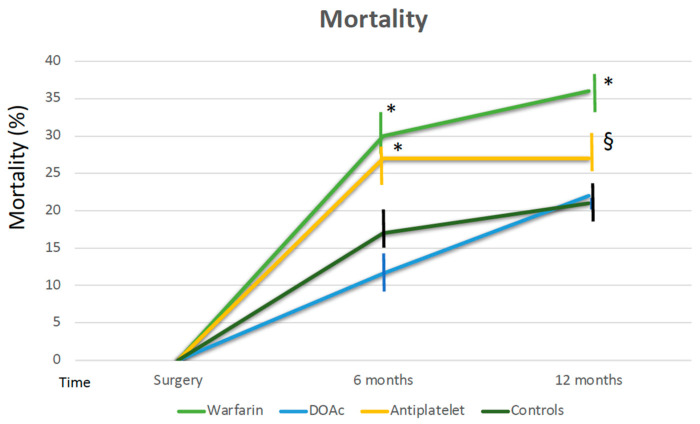
Mortality at 6 and 12 months according to therapy. * *p* < 0.0001 warfarin and P2Y12 inhibitors vs. controls and DOAc at 6 months, * *p* < 0.0001 warfarin vs. controls and DOAC at 12 months, § *p* = 0.02 P2Y12 inhibitors vs. controls and DOAC at 12 months.

**Figure 3 jcm-12-06178-f003:**
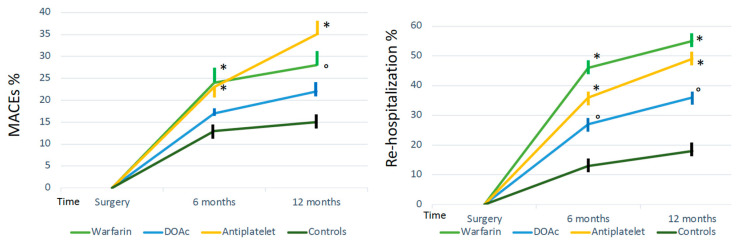
Major cardiovascular events (MACEs) and re-hospitalization rate according to therapy. MACEs * *p* = 0.005 warfarin and P2Y12 inhibitors vs. controls and DOAc at 6 months, * *p* = 0.005 warfarin vs. controls and DOAC at 12 months, ° *p* = 0.015 P2Y12 inhibitors vs. controls and DOAc at 12 months. Re-hospitalization * *p* < 0.0001 warfarin and P2Y12 inhibitors vs. controls and DOAc at 6 months, * *p* < 0.0001 warfarin and P2Y12 vs. controls and DOAC at 12 months, ° *p* = 0.02 DOAc vs. controls at 6 and 12 months.

**Table 1 jcm-12-06178-t001:** Characteristics of patients.

Characteristics	Patients n = 138	Controls n = 283	*p*
Age (±SD)	85.9 (±6.3)	86.1(±5.7)	0.8
Female (%)	104 (75.4%)	218 (77%)	0.6
Basic Activity of Daily Living (preserved ≤ 4)	40 (29%)	96 (33%)	0.4
Time to surgery (days)	2.95 (±1.66)	1.8 (±1.3)	<0.0001
Treatment Surgery within 48 h (%)	50%	82%	<0.0001
Length of Stay (LOS), days	12.6 (± 3.93)	11.5 (±2.7)	0.009

**Table 2 jcm-12-06178-t002:** Indications of antithrombotic treatment.

Clinical Indication	Warfarin	DOAc	P2Y12 Inhibitors
Atrial fibrillation	26	70	4
Valvular pathology	4	1	0
Deep vein thrombosis and pulmonary embolism	3	5	1
Other	4	6	15

**Table 3 jcm-12-06178-t003:** Characteristics of patients stratified by therapy.

Characteristics	Warfarin (*n* = 36)	DOAc (*n* = 82)	P2Y12 Inh. (*n* = 20)	Controls (*n* = 283)	*p*
Age	87.3 ± 7.3	86.7 ± 5.3	86.1 ± 3.8	86.1 ± 5.7	0.7
Female (%)	27 (75%)	62 (75.6%)	15 (75%)	218 (77%)	0.7
BADL ≤ 4	13%	39%	15%	33%	0.03
Type of fracture					
Femoral neck	16	35	5	99	0.008
Pertrochanteric	16	45	9	143	
Subtrochanteric	4	2	4	43	
Time to surgery	3.3 ± 1.7	2.8 ± 1.6	2.8 ± 1.5	1.8 ± 1.3	<0.0001
Surgery < 48 h	38%	53%	55%	82%	<0.0001
GFR	53 ± 36	58 ± 30	56 ± 19	58.6 ± 13	0.5
Length of hospital stay (LOS), days	13.2 ± 2.9	12.1 ± 4.3	13.5 ± 3.4	11.5 ± 2.7	0.0011

**Table 4 jcm-12-06178-t004:** Comparison of early and late surgery mortality at 6 and 12 months.

	Surgery < 48 h *n* = 79	Surgery > 48 h *n* = 79	*p*
Warfarin			
In hospital	0	0	0.8
6 months	30%	34%	0.5
12 months	38%	39%	0.7
DOAC			
In hospital	0	3%	ns
6 months	9.5%	14%	0.3
12 months	18%	24%	0.2
P2Y12 inhibitors			
In hospital	0	0	0.8
6 months	36%	12.5%	0.09
12 months	36%	12.5%	0.09

## Data Availability

Data are available on request.

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
