# Peer review of "Management and 1-Year Outcome in Elderly Patients with Hip Fracture Surgery Receiving Anticoagulation (Warfarin or DOAc) or P2Y12 Antiplatelet Agents"

_jcm, 2023, doi:10.3390/jcm12196178_

Round 1
Reviewer 1 Report
Review
Many thanks to the Authors for having presented a so interesting original article about
“Management and 1 Year Outcome in Elderly Patients with Hip Fracture Surgery Receiving
Anticoagulation or P2Y12 Antiplatelet Agents”.
The article is written mostly in a correct English. However, there are multiple grammatical,
punctuation and syntactic errors throughout the text that severely undermine the comprehension of
the concepts that the Authors are trying to explain. Hence the manuscript needs to be corrected by a
native English speaker.
Throughout the article, the Authors refer to the medications as anticoagulant/DOAC/DOAc and
warfarin/VKA/antivitamin K. We strongly recommend the Authors to use a single definition for
each medication.
Moreover, throughout the article, the Authors write numerical values (such as percentages, number
of patients enrolled in the study, etc.) in letters or in numerals with no consistency. We suggest the
Authors to write numerical values in numeral for a easier and immediate comprehension.
Title and Abstract
In the title, it is specified that the cohort of patients of the study are in Anticoagulation OR P2Y12
Antiplatelet therapy. However, throughout the article, multiple times is written that the patients are
in Anticoagulation AND/OR P2Y12 Antiplatelet therapy. We think the Authors should clarify or
correct their statements.
In the abstract are listed the end points of the study. In other parts of the article (Introduction, Study
Design and Study Population), the end points listed vary, with one or more being added or
subtracted without consistency. We suggest the Authors to agree on a coherent list of end points.
2
Key words
The keywords should be provided in an alphabetical order and should include words strictly related
to the scope of the study.
Introduction
The introduction provides background and information relevant to the study and it identifies the
problem that is being address in the manuscript. Further, in this section the purpose of the
manuscript is clearly developed and stated.
Line 42: In hip fracture early surgery, within 24– 48 h from trauma, is associated with better
outcomes both in term of recovery of functional capacity and overall early and 1-year survival [2,3].
Please, add the importance of the correct management of these patients during the hospitalisation
and quote:
Efficacy of an interdisciplinary pathway in a first level trauma center orthopaedic unit: A
prospective study of a cohort of elderly patients with hip fractures. Arch Gerontol Geriatr.
2020 Jan-Feb;86:103957. doi: 10.1016/j.archger.2019.103957. Epub 2019 Oct 12.
Methods
At line 76 the Authors cite the approval of the project by the ethical committee of February 19 th
2019, n° 6590/2109. We suppose that the last number should refer to the year (2019).
In the Study Design, at line 77, the Authors write that the patient OR their legal guardians provided
the informed consent to the procedure and the enrollment in the study. However, later at line 84, the
3
Authors affirm that one of the exclusion criteria is the ability form the patient to actively provide
the informed consent. We believe that the Authors contradicted themselves.
At line 86, one of the exclusion criteria is patient in double antiplatelet therapy for recent coronary
revascularization. The Authors should specify the time frame.
Statistical analysis
Please provide who performed the analysis: an independent statistician or the same Authors?
Results
The results presented are quite complete, reflecting the MM section.
At line 142, the Authors cites Table 2 while referring to results listed in Table 1.
Discussion
The length and content of the discussion communicates the main information of the paper.
Study Limitations
This study is based on patients with fragility hip fractures. The patients received different surgical
treatments, such as nailing or endo/arthroplasty. The results obtained from their observation were
4
not differentiated based on the type of surgery provided. Do the Authors see this as limitation to
their study?
Conclusions
The conclusions only reflect and refer to the results of the study.
References
The references are up to date, but they should be integrated as suggested previously.
Tables and Figures
The colors and shape of the lines of the results in Figures 2 and 3 are easily confused. Please,
rewrite them. Table 4 has no description: please, provide proper legend.
Minor editing of English language required
Author Response
Many thanks to the Authors for having presented a so interesting original article about
“Management and 1 Year Outcome in Elderly Patients with Hip Fracture Surgery Receiving
Anticoagulation or P2Y12 Antiplatelet Agents”.
The article is written mostly in a correct English. However, there are multiple grammatical,
punctuation and syntactic errors throughout the text that severely undermine the comprehension of
the concepts that the Authors are trying to explain. Hence the manuscript needs to be corrected by a
native English speaker.
Throughout the article, the Authors refer to the medications as anticoagulant/DOAC/DOAc and
warfarin/VKA/antivitamin K. We strongly recommend the Authors to use a single definition for
each medication. We agree with your observation and modify the paper according to suggestions
Moreover, throughout the article, the Authors write numerical values (such as percentages, number
of patients enrolled in the study, etc.) in letters or in numerals with no consistency. We suggest the
Authors to write numerical values in numeral for a easier and immediate comprehension. Again we changed the text and we used numerical values
Title and Abstract
In the title, it is specified that the cohort of patients of the study are in Anticoagulation OR P2Y12
Antiplatelet therapy. However, throughout the article, multiple times is written that the patients are
in Anticoagulation AND/OR P2Y12 Antiplatelet therapy. We think the Authors should clarify or
correct their statements. Patients were in anticoagulants (warfarin or DOAc ) or P2Y12 inhibitors, this in now well specified in the text .
In the abstract are listed the end points of the study. In other parts of the article (Introduction, Study
Design and Study Population), the end points listed vary, with one or more being added or
subtracted without consistency. We suggest the Authors to agree on a coherent list of end points.
2
Key words
The keywords should be provided in an alphabetical order and should include words strictly related
to the scope of the study. The key words were modified and inserted in alphabetical order
Introduction
The introduction provides background and information relevant to the study and it identifies the
problem that is being address in the manuscript. Further, in this section the purpose of the
manuscript is clearly developed and stated.
Line 42: In hip fracture early surgery, within 24– 48 h from trauma, is associated with better
outcomes both in term of recovery of functional capacity and overall early and 1-year survival [2,3].
Please, add the importance of the correct management of these patients during the hospitalisation
and quote:
- Efficacy of an interdisciplinary pathway in a first level trauma center orthopaedic unit: A
prospective study of a cohort of elderly patients with hip fractures. Arch Gerontol Geriatr.
2020 Jan-Feb;86:103957. doi: 10.1016/j.archger.2019.103957. Epub
2019 Oct 12. The sentence was added as well the reference added
Methods
At line 76 the Authors cite the approval of the project by the ethical committee of February 19 th
2019, n° 6590/2109. We suppose that the last number should refer to the year (2019). The text was modified
In the Study Design, at line 77, the Authors write that the patient OR their legal guardians provided
the informed consent to the procedure and the enrollment in the study. However, later at line 84, the
3
Authors affirm that one of the exclusion criteria is the ability form the patient to actively provide
the informed consent. We believe that the Authors contradicted themselves. Only patients who gave informed consent were included in the study. The text was corrected.
At line 86, one of the exclusion criteria is patient in double antiplatelet therapy for recent coronary
revascularization. The Authors should specify the time frame. Patients who underwent revascularization within 6 months from trauma were excluded from the study
Statistical analysis
Please provide who performed the analysis: an independent statistician or the same Authors? Statistical analysis was performed by the authors
Results
The results presented are quite complete, reflecting the MM section.
At line 142, the Authors cites Table 2 while referring to results listed in Table 1. The number was changed
Discussion
The length and content of the discussion communicates the main information of the paper. Thank you
Study Limitations
This study is based on patients with fragility hip fractures. The patients received different surgical
treatments, such as nailing or endo/arthroplasty. The results obtained from their observation were
4
not differentiated based on the type of surgery provided. Do the Authors see this as limitation to
their study? A sub analysis did not show differences regarding the type of surgery, however the number of patients is limited to draw definite conclusion. It was added to limitations
Conclusions
The conclusions only reflect and refer to the results of the study. The paragraph was changed according to your suggestions
References
The references are up to date, but they should be integrated as suggested previously.
Tables and Figures
The colors and shape of the lines of the results in Figures 2 and 3 are easily confused. Please,
rewrite them. Table 4 has no description: please, provide proper legend. Changes were made
Reviewer 2 Report
I read with great interest the manuscript by Rostagno et al. on anticoagulation or P2Y12 antiplatelet agents e in elderly patients undergoing hip fracture surgery. The article is sound and well written. However, there are some issues to be addressed:
- Line 44. Authors should also add that several strategies involving loco-regional anesthesia have been studied to improve outcomes of these patients (doi: 10.1186/s44158-022-00047-6 - doi: 10.12688/f1000research.130387.2). Please briefly discuss and add these 2 references.
- Line 53-55. It is also important to underline that new anticoagulation drugs are being tested in recent years (doi: 10.7759/cureus.23020 - doi: 10.1111/aor.14276 - doi: 10.3390/jcm12154984), thus requiring further studies on their long term effects. Please briefly discuss and add these 3 references.
- Line 79. Please report and cite the relevant guidelines mentioned.
- Line 86. Number of patients recruited should be presented in the results section. Please modify.
- Line 92. Even this information should be reported in the methods section.
-Line 96. Please replace "i" before the word "warfarin".
- Table 1, 3 and 4. Please report p values even though they are not significant.
- Line 212-213. Please correct errors of syntax.
- Please report the observational design of the study as a limitation.
Author Response
I read with great interest the manuscript by Rostagno et al. on anticoagulation or P2Y12 antiplatelet agents e in elderly patients undergoing hip fracture surgery. The article is sound and well written. However, there are some issues to be addressed:
- Line 44. Authors should also add that several strategies involving loco-regional anesthesia have been studied to improve outcomes of these patients (doi: 10.1186/s44158-022-00047-6 - doi: 10.12688/f1000research.130387.2). Please briefly discuss and add these 2 references. The references were added and discussed
- Line 53-55. It is also important to underline that new anticoagulation drugs are being tested in recent years (doi: 10.7759/cureus.23020 - doi: 10.1111/aor.14276 - doi: 10.3390/jcm12154984), thus requiring further studies on their long term effects. Please briefly discuss and add these 3 references. The references were added and discussed
- Line 79. Please report and cite the relevant guidelines mentioned. Made
- Line 86. Number of patients recruited should be presented in the results section. Please modify. Made
- Line 92. Even this information should be reported in the methods section. Made
-Line 96. Please replace "i" before the word "warfarin". Made
- Table 1, 3 and 4. Please report p values even though they are not significant. P values were reported in each table
- Line 212-213. Please correct errors of syntax. Made
- Please report the observational design of the study as a limitation. The sentence was added
Reviewer 3 Report
The authors focus on the management and one-year outcome in elderly patients with hip fracture surgery receiving anticoagulation or P2Y12 antiplatelet agents. The topic is very interesting, but some things have to be corrected. The discussion was carried out correctly. The presented conclusions are consistent. References typical and actual.
Shortcomings:
- The paper is hard to read. All English language should be modified. Multiple spelling, format, and punctuation errors should be corrected, like line 22, 60, 66, 74, 82, 96, 124 etc.
- Check the keywords section.
- All of the tables and graphs have to be described.
- Indicate the SEM in all graphs and tables.
- All graphs should show the units of the Cartesian axes.
- Always indicate towards which group significance is intended.
- There is a lack of reference in line 259
- There is no mention of references 14-18 in the text.
- The reference format should be adjusted to the publisher's requirements.
The paper is hard to read. All English language should be modified.
Author Response
REVIEWER 3
The authors focus on the management and one-year outcome in elderly patients with hip fracture surgery receiving anticoagulation or P2Y12 antiplatelet agents. The topic is very interesting, but some things have to be corrected. The discussion was carried out correctly. The presented conclusions are consistent. References typical and actual.
Shortcomings:
- The paper is hard to read. All English language should be modified. Multiple spelling, format, and punctuation errors should be corrected, like line 22, 60, 66, 74, 82, 96, 124 etc. The text has been extensively revised
- Check the keywords section. Made
- All of the tables and graphs have to be described. Made
- Indicate the SEM in all graphs and tables. Made
- All graphs should show the units of the Cartesian axes. Made
- Always indicate towards which group significance is intended. Made
- There is a lack of reference in line 259 Inserted
- There is no mention of references 14-18 in the text. Inserted
- The reference format should be adjusted to the publisher's requirements. Made
Round 2
Reviewer 3 Report
The authors of this manuscript focus on management and one-year outcome in elderly patients with hip fracture surgery receiving anticoagulation or P2Y12 antiplatelet agents.
The study is very interesting, and the manuscript is well-written after the revision. This article may be eligible for publication after correcting minor noticed errors.
- Remove the word “author” – line: 12.
- There are still a lot of punctuation errors - lines: 12, 19, 23, 25, 26, 27, 28, 29, 50, 54, 55, 56, 60, 63, 66, 75, 85, 90, 100, 110, 121, 141, 142, 148, 149, 166, 181, 187, 211, 217, 225, etc.
- Add “n=138” in Table 1 and the description of “n” and “p” below Table 1.
- Add the description of numbers in Figure 1.
- Add the description of numbers in Table 2.
- There is lack of “=” in the header of table 3, there is lack of “-“ in “±” (GFR controls) in the same table.
- Table naming in the text is inconsistent (table 1), (Table 2) interchangeable.
- There is a lack of the title in Table 4 and the header in columns 1 and 5.
- Graphs in Figures 2 and 3 are illegible.
- Description of “p” differs in the mortality graph in figure3. The same description is pasted in line 196.
- All graphs should show the units of the Cartesian axes.
- Indicate towards which group significance is intended.
- The style of references should be adapted to the requirements of the publishing house – see template.
A lot of punctuation errors.
Author Response
Dear Sir ,
please find below the answer to your questions. We tried to improve the paper and we hope that now it may be suitable for publication.
Kind regards
The authors of this manuscript focus on management and one-year outcome in elderly patients with hip fracture surgery receiving anticoagulation or P2Y12 antiplatelet agents.
The study is very interesting, and the manuscript is well-written after the revision. This article may be eligible for publication after correcting minor noticed errors.
- Remove the word “author” – line: 12. The word was removed
- There are still a lot of punctuation errors - lines: 12, 19, 23, 25, 26, 27, 28, 29, 50, 54, 55, 56, 60, 63, 66, 75, 85, 90, 100, 110, 121, 141, 142, 148, 149, 166, 181, 187, 211, 217, 225, etc. Punctuation errors were checked and corrected
- Add “n=138” in Table 1 and the description of “n” and “p” below Table 1. Done
- Add the description of numbers in Figure 1. The description was added
- Add the description of numbers in Table 2. The description is above the table
- There is lack of “=” in the header of table 3, there is lack of “-“ in “±” (GFR controls) in the same table. Added
- Table naming in the text is inconsisteMadent (table 1), (Table 2) interchangeable. Corrected
- There is a lack of the title in Table 4 and the header in columns 1 and 5. Added
- Graphs in Figures 2 and 3 are illegible. Graphs were remade. Figure 3 was deleted and figure 3 was divided in two new figures.
- Description of “p” differs in the mortality graph in figure3. The same description is pasted in line 196. Corrected
- All graphs should show the units of the Cartesian axes. Done
- Indicate towards which group significance is intended. Reported in the legend
- The style of references should be adapted to the requirements of the publishing house – see template.
Done